# Effects of Flight Restraint and Housing Conditions on Feather Corticosterone in White Storks Under Human Care

**DOI:** 10.3390/ani15131878

**Published:** 2025-06-25

**Authors:** Frederike Liermann, Katrin Baumgartner, Ralph Simon, Hermann Will, Lorenzo von Fersen, Roswitha Merle, Christa Thöne-Reineke

**Affiliations:** 1Animal Behaviour and Laboratory Animal Science, Institute of Animal Welfare, Freie Universität Berlin, Königsweg 67, D-14163 Berlin, Germany; 2Zoo Nuremberg, Am Tiergarten 30, D-90480 Nuremberg, Germany; 3Machine Learning and Data Analytics Lab, Department Artificial Intelligence in Biomedical Engineering (AIBE), Friedrich-Alexander-Universität Erlangen-Nürnberg (FAU), Carl-Thiersch-Straße 2b, D-91052 Erlangen, Germany; 4Institute of Veterinary Epidemiology and Biostatistics, Freie Universität Berlin, Königsweg 67, D-14163 Berlin, Germany

**Keywords:** white stork, feather corticosterone, animal welfare, flight restraint, zoological institutions

## Abstract

Flight restraint in zoo birds raises welfare concerns, yet scientific data on its impact remains limited for many species. Our study examined whether flight restraint affected welfare in white storks by determining stress hormones (corticosterone) in feathers and observing behaviour. We compared fifty-three flight-restrained zoo birds, eleven hand-raised nestlings from German zoos, and seventy wild storks in rehabilitation. Using a minimally invasive sampling method that only required cutting feathers close to the skin, we tested whether flying storks would show different stress hormone levels compared to flight-restrained individuals. Surprisingly, we found no significant differences in hormone levels related to flight ability or restraint method. Instead, the location where storks were kept emerged as the main factor influencing stress hormone levels, with significant variations between different sites. Our behavioural observations supported these findings. This study provides valuable evidence that flight restraint may not directly impact stress levels in white storks, contributing important information for zoo management practices and avian welfare assessment in settings under human care.

## 1. Introduction

The welfare of deflighted birds in zoological institutions is gaining increasing attention. The practice of flight restriction, while often implemented for safety and management purposes, may lead to significant welfare challenges for these birds [1]. Research has shown that depriving birds of their natural flight behaviours could result in stress, decreased physical fitness, and the development of stereotypic behaviours. However, other studies demonstrated no effect on their behaviour or stress levels [2,3,4,5,6]. As public awareness of animal welfare continues to grow, and as zoos are committed to the welfare of their animals, it is essential to critically examine the implications of deflighted conditions in birds under human care and to explore strategies for enhancing their care and living environments [6,7,8,9]. Zoological gardens comply with the European Union’s Zoos Directive (1992/22/EC), which mandates that the biological and conservation needs of each species are addressed by tailored enrichment and proper husbandry practices [10]. This is particularly relevant for large wading birds such as white storks, which are often housed in open enclosures where technical or legal barriers prevent full containment. In such settings, flight restraint becomes a common, yet controversial, management tool [6,7,8,9,11,12].

The most common methods for flight restraint in birds can be categorised into reversible techniques, such as feather clipping (trimming the primary feathers on one side), and irreversible procedures, such as pinioning as the most common irreversible deflighting method. Pinioning describes the surgical removal of the distal portion of the wing, resulting in the amputation site located at the metacarpals below the alula in most cases [6,8,12]. Feather follicle management includes excision of primary feather follicles by surgical removal (extirpation) or destruction of germinal tissue using a diode laser or cryoprobe via the feather shaft, methods occasionally employed in Germany in recent decades [12]. Patagiectomy, an uncommon irreversible method used in large birds, prevents full wing extension by removing part of the patagial membrane [6]. While §6 of the German Animal Welfare Law now strictly prohibits irreversible methods like pinioning [13], such procedures remain legal or tolerated in parts of Europe. The legal framework governing the practice of rendering birds flightless under human care varies considerably between countries and even among municipalities. For example, irreversible methods are prohibited throughout Germany and reversible methods are generally banned as well. In some European countries, irreversible deflighting methods remain legally permitted, or exemptions are granted at the municipal level [12]. There is the opinion that flight restraint procedures could be an effective means to balance animal welfare and management goals for specific bird species in certain cases [1,6]. Conversely, other researchers oppose such practices, viewing them as outdated or ethically unacceptable, with some even advocating for their prohibition [5,14]. To date, no clear consensus on their welfare implications has been reached [1,6,15].

How to assess animal welfare correctly? Animal welfare is defined as an animal’s ability to cope with its environment, encompassing both physical and psychological health [16]. This definition emphasises the importance of assessing welfare objectively, including both the animals’ physical and psychological health as well as their emotional experiences, while avoiding reliance on purely moral considerations [17]. Objective assessments commonly rely on behavioural observations, endocrine markers, and physical condition [18,19,20]. Behaviour alone may not reflect internal stress states; therefore, physiological measures like corticosterone, the main avian glucocorticoid, offer a more robust indicator of chronic stress and allostatic load [16,18,19,20,21,22,23,24,25,26]. Research suggests that elevated corticosterone levels may not solely indicate stress or poor welfare but can also reflect adaptive physiological responses to enriched environments, potentially associated with positive welfare outcomes [27], making this hormone a valuable biomarker in welfare research on white storks [28,29,30,31,32].

Non-invasive methods to assess glucocorticoids are increasingly used to overcome the limitations of plasma-based analyses, which only reflect acute stress and are affected by handling and haemodilution [33]. In birds, salivary measures have proven impractical [4], while faecal glucocorticoid metabolites offer integrated values over time, considering species-specific gut transit [34]. Aligned with Directive 2010/63/EU and the 3Rs (Replacement, Reduction, Refinement) [35], feather corticosterone was used here as a non-invasive, validated method involving cutting feathers close to the skin [36]. Samples must be morphologically consistent and free of contamination [37,38,39]. CORT integrates into keratin during feather growth, providing a long-term reconstruction of hormone exposure [38] and allowing a retrospective view on the moulting period of the white stork. White storks undergo a staggered moulting cycle over one to two years, making it possible to retrospectively evaluate hormonal status during specific periods [40,41]. This study was conducted during spring and summer, coinciding with breeding, moulting, and the intake of rescued birds, allowing broad data collection across groups.

For many bird species, it is still not clear whether commonly deflighted bird species experience lower welfare levels or whether the ability to fly is nonessential or outweighed by other behaviours deemed more critical [12,42]. Few studies have examined the impact of flight restriction on welfare in a species-specific context. Reese et al. analysed CORTf in greater flamingos, and Haase et al. expanded this to great white pelicans (*Pelecanus onocrotalus*) [2,3]. Both studies concluded that the inability of birds to fly is not inherently the cause of reduced welfare; instead, housing conditions such as enclosure design and group size play a major role. These findings suggest that broader environmental and management factors must be considered when assessing welfare in deflighted birds.

White storks (*Ciconia ciconia*) are a relevant model species for this purpose. They are commonly kept under flight restraint in open zoo enclosures, and their biology and behaviour in the wild are well documented [43,44,45,46]. As adept long-distance fliers that can cover thousands of kilometres during migration [47,48,49,50], they provide a meaningful model to evaluate how flight (in)ability affects stress and welfare under human care. In response to growing welfare concerns, the Veterinary Association for Animal Protection (Tierärztliche Vereinigung für Tierschutz—TVT) points out the necessity of advancing ethological and physiological research on white storks to better understand and optimise various husbandry practices [9].

This study aimed to compare feather corticosterone (CORTf) levels across different stork populations: wild, flight-capable birds and zoo populations subjected to varying flight restraint methods. We hypothesised that CORTf levels would be significantly higher in deflighted storks, indicating increased allostatic load and stress. In addition, we examined how housing and husbandry factors influenced both hormonal and behavioural welfare indicators.

## 2. Materials and Methods

### 2.1. Bird Numbers

This study included deflighted white storks housed in zoos under flight restraint, hand-reared nestlings, as well as injured wild white storks in rehabilitation centres. Eleven zoos affiliated with the German Association of Zoological Gardens (VdZ) participated, and ten allowed video recordings. In three zoos, wild, airworthy storks mingled with deflighted individuals (locations C, G, and I), and only flight-incapable individuals were sampled. Additional data was obtained from two rehabilitation centres housing injured, non-airworthy wild storks. Samples from juvenile nestlings were excluded from analysis due to uncertainty regarding flight status, and their data is presented exclusively in the boxplots.

The sample size for this study was determined based on practical considerations. The key factors influencing the sample size included the availability of animals and zoos; logistical constraints such as financial resources, empirical evidence and prior studies [2]; and animal welfare considerations, particularly adhering to the 3R principle to minimise the use of animals and ensure ethical research practices. To account for the expected cluster effect (intraclass correlation coefficient (ICC) = 0.493) within the zoos, the number of animals per zoo was estimated based on preliminary data [2]. In cases where a zoo housed fewer than 10 individuals, all animals were included in the sample. The number of sampled animals per zoo can be seen in Table 1.

Consequently, feathers were cut from 10 randomly selected storks in each participating zoo with 10 or more individuals: The animals were selected randomly by the observers, without consideration of their position in the enclosure or individual characteristics.

### 2.2. Study Design

#### 2.2.1. Behavioural Observations

In ten of the participating zoos, camera recordings were conducted between April and August 2022. This time frame was chosen for management reasons, as all zoo birds were kept in outdoor enclosures, contributing to standardised experimental conditions. One zoological institution, though, housed a wild population, and a breeding pair was filmed for practical reasons. Additionally, the chosen period coincided with wild, migratory birds still being in northern latitudes and their nestlings hatching [51].

Questionnaire surveys were developed in collaboration with a statistician and veterinarian, based on preliminary studies executed under comparable conditions (the full survey is available in the Appendix A) [3]. During the 2022 sampling period, these surveys were completed by responsible veterinarians, curators, and zookeepers to gather and categorise information on white stork management. The ten questions as part of the questionnaire focused on topics such as enclosure size, flight restraint methods, group size, feeding practices (frequency, type, and quantity), health status (acute vs. chronic diseases), and cohabitation with other species.

In all zoos, data collection was carried out on three consecutive days between 07:00 a.m. and 07:00 p.m., with filming taking place for ten minutes at each full hour over eight hours daily.

Instantaneous scan sampling was employed to assess the relative distribution of behaviours within the flock for the qualitative behavioural assessment [52]. Scans always included the whole flock. A detailed ethogram was created and applied based on observations of white storks from the Nuremberg Zoo and the cited literature [1] and can be found in Table 2. Based on this previously established ethogram, white storks’ behaviour was documented in person using scan sampling for 1.5 h in one zoo and two hours in the second. All behavioural footage was recorded with a custom-build programmable CoSysCam camera (CoSys Lab, Antwerp University, Antwerp, Belgium) or GoPro7 (GoPro, San Mateo, CA, USA). Each video clip from each facility was analysed in two-minute intervals, recording the number of individuals displaying specific behaviours. The total occurrences of each behaviour were converted into percentages, resulting in a behavioural profile for each zoological institution, summing up to 100%.

#### 2.2.2. Feather Sampling

To align with animal welfare considerations, a representative sample size was selected for each group, including factors such as colony size, flight status, and study group. Considering the standardised housing of white storks in outdoor enclosures during summer and the concurrent behavioural observations, this period was selected for sampling. To ensure consistency between conditions, all feather samples were collected between February and August 2022. This study followed the 3R principle developed by Russell et al. using the method of Voit et al. [36]. Feather samples were obtained during routine treatments of white storks, such as removing them from winter quarters, banding the nestlings, or conducting medical examinations on injured wild storks to minimise additional stress. This technique, chosen as the preferred method in this study, did not require an animal experimentation permit as it was not classified as such.

A total feather shaft length of 200 mm per individual was required. As Carbajal et al. specifies, four to five feathers, approximately 50 mm in length, were cut as close as possible to the skin between the shoulder blades and stored in tightly sealed, dry envelopes at room temperature [53,54]. A total of 64 feather samples were collected from zoo-housed white storks, including 11 samples from hand-reared zoo nestlings categorised as potentially flight-capable, while the remaining 53 individuals of this group were considered incapable of flight. Additionally, 70 feather samples were obtained from wild white storks that had suffered injuries. Sixty-two of these birds were definitively flight-capable during feather growth. However, ten of these sixty-two flight-capable storks had been in human care for at least six months. As a result, their data did not contribute to the category of flight-capable wild storks but rather to the category of flight-capable storks kept under human care. Therefore, they were assigned to the valuable control group of the flightless zoo-housed storks (see Result section for visualisation). We divided the group of rehabilitated storks into three categories. Category 1 included individuals that were sampled shortly after their accident, thus representing values from their flight-capable period in the wild. Category 2 consisted of storks that had lived under human care for at least six months until one and a half year, while Category 3 included those that had been in human care for several years. This final categorisation was implemented to investigate whether prolonged moult progression under human care conditions correlated with variations in corticosterone concentrations.

### 2.3. Corticosterone Extraction and Measurement

Corticosterone (CORTf) was extracted from feather samples following the method described by Bortolotti et al. (2008) and refined by Monclús et al. (2017) [54,55]. The prepared samples were analysed using an enzyme-linked immunosorbent assay (ELISA).

A uniform feather powder was created out of 0.1 mg using a ball mill (Retsch^®^, MM200 type with two balls at 25 Hz, Haan, Germany) to achieve 10 µm particles and re-weighed to ensure minimal material loss. Each sample was mixed with 1.5 mL of methanol in a vortex mixer (Vortex Mixer S0200-230 V-EU; Labnet International, Edison, NJ, USA) for 30 min at room temperature. The mixture then underwent an 18 h incubation for steroid extraction in a shaking incubator (G24 Environmental Incubation Shaker, New Brunswick Scientific, Edison, NJ, USA) at 37 °C. After centrifugation for 15 min at 3500× *g* (Hermle Z300K; Hermle^®^ Labortechnik, Wehingen, Germany), 1 mL of the supernatant was transferred to an Eppendorf^®^ reaction tube. This sample was dried for several hours at 30 °C in an oven (Heraeus Function Line T6^®^, Thermo Fisher Scientific, Waltham, MA, USA). The residue was dissolved in 250 µL of buffer solution by vortexing. The buffer solution, compatible with an ELISA kit (ELISA Neogen^®^ Corporation, Ayr, UK), contained BSA, NaCl, EDTA, and Azide. If there was a delay after processing, the sample was frozen at −20 °C until analysis.

It was assumed that feather growth rates would be consistent among morphologically similar feathers from the same individual and location [38]. Therefore, the CORTf value obtained from the ELISA were reported in pg/mm of feather length. The interpretation of the CORTf value is dependent on the specific feather length used, reflecting the hormone exposure over time [38,55,56].

### 2.4. Statistical Analysis

Statistical analyses were performed using R version 4.3.2 and R-Studio version 2023.09.1+494 with the lme4, performance, and emmeans packages. First, the Shapiro–Wilk test was applied to assess the normal distribution of continuous values (CORTf). Due to the non-normal distribution of CORTf, logarithmic transformations to the base of 10 were conducted to normalise the data. Behavioural data from video recordings was expressed as percentages. Model assumptions including normality of residuals were assessed using Q-Q plots.

Univariable analyses were conducted testing each of the following fixed effects individually: flight status, location, study group, individual weight, rehabilitation category, time of sampling, and deflighting method (Table 3). Variables that demonstrated significance in the univariable analyses (*p* < 0.05) were subsequently included in a multivariable analysis. To determine the optimal model structure, models were fitted using the lmer function from the lme4 package in R. Model selection was based on Akaike’s Information Criterion (AIC), where lower values indicate better model fit, as well as assessment of R^2^ values to determine the variance explained by fixed effects (marginal R^2^) versus combined fixed and random effects (conditional R^2^). After comparing models with different random effect structures, we determined that treating location, rehabilitation category, study group, and flight status as fixed effects and no random effect provided the best model fit (Table 4). The final model included rehabilitation category, study group, and location as fixed effects (Table 4). Flight status did not significantly improve model performance based on AIC values and likelihood ratio tests (ANOVA), though. However, we retained it in the model due to its relevance to our hypothesis. Post hoc multiple comparisons using Tukey’s method were conducted to find significant differences between several rehabilitation categories and locations. Statistical significance was defined as *p* < 0.05.

## 3. Results

The baseline log-transformed corticosterone levels (logCORTf) in wild white storks (*n* = 70) were found to be 3.81 pg/mm (CORTf 45.08 pg/mm). In the sampled population of zoo birds (*n* = 53, without zoo nestlings), the logCORTf values ranged from a minimum of 2.31 pg/mm to a maximum of 5.37 pg/mm (simple CORTf: min 10.05 pg/mm, max = 215.10 pg/mm). The median CORTf value for this group was calculated to be 39.67 pg/mm, with a corresponding median logCORTf of 3.68 pg/mm (Figure 1).

The final multivariable linear model incorporated all univariable factors that could potentially influence corticosterone levels (see Table 3). After excluding variables and their interactions, the final model retained the following predictors: location (zoological institutions), flight status, study group, and rehabilitation category (see Table 4 and Figure 2). The final model included these four factors as fixed effects, with rehabilitation category being the only factor showing statistical significance in univariable analysis (*p* < 0.01). Flight status (*p* = 0.058) and location (zoological institution/rehabilitation centres) were not significant. 

Numerically higher log CORTf values were observed for flight-capable storks compared to incapable ones (mean: 3.86 vs. 3.64). However, the difference was not statistically significant (t = 1.92, *p* = 0.058, 95% CI: [−0.007, 0.461]). Statistical analysis using univariable linear mixed effects models revealed significant effects of rehabilitation category (F (2, 65.06) = 10.63, *p* < 0.001), study group (F (2, 29.67) = 14.67, *p* < 0.001), and location (F (12, 11.12) = 10.42, *p* < 0.001) on log CORTf values. “Rehabilitation category 1” served as the reference level in our analysis (Figure 3). The multivariable model comparison using AIC values confirmed that the reduced model without flight capability as a predictor provided a better fit to the data. The estimates from our models indicate that storks in rehabilitation categories 2 and 3, as well as zoo-housed storks and hand-raised nestlings, exhibited significantly lower corticosterone values compared to the reference intercept.

The observed behavioural patterns exhibited considerable variability in their relative frequencies, as Figure 4 shows. The high levels of variability in the observed behavioural patterns were attributed to differences in breeding cycles, group sizes, enclosure structures, and cohabitation with other species. The behavioural recordings did not capture any flying storks in the three facilities with a mixed population during the filmed time frames, so flying was not included in the ethogram. Moreover, in two of the zoological institutions, pairs were exclusively observed during breeding (N) or rearing (K), while in another (G), only a small portion of the overall group was involved in reproduction. No aggressive behaviour was observed in animals held as pairs (A, B, E, K), whereas such behaviour tended to be more common in larger groups (C, G, H). However, there were also groups (F, J) where no such behaviour was recorded. Statistical analysis revealed no significant association between the behaviour of fluttering and elevated corticosterone levels (*p* > 0.05). Additionally, storks with restricted flight did not exhibit high locomotion scores, and groups with increased movement did not show higher logCORTf values.

## 4. Discussion

Our main finding revealed no significant impact of flight restraint on corticosterone levels in white storks. Importantly, behavioural observations did not undermine the reliability of these results. The following section highlights key results from the statistical analysis to be discussed here.

This study included two distinct groups to test our hypothesis: wild, airworthy white storks and deflighted white storks residing in zoos. It is important to note that nestlings are shown in the boxplots as flight-capable zoo nestlings; however, their data must be interpreted with caution, as their flight (in)capacity is part of a natural developmental stage and does not reflect any restriction that could influence corticosterone levels.

Comparing wild and flight-restrained white storks revealed the following differences: Wild individuals showed marginally numerically higher CORTf levels than zoo-housed storks. This is in contrast to other studies reporting higher glucocorticoid (GC) levels in captive animals, such as Karaer’s study on mammals (such as lynx, lemur, cheetah and spider monkey) (2023) [57] and research on wild spider monkeys (*Ateles geoffroyi yucatanensis*) by Rangel-Negrín et al. (2009) [58]. However, a conclusion cannot be drawn due to differences in class and species selection and analytical approaches. A possible explanation for lower CORTf values in zoo birds could be the exposure to chronic stress, which has previously been shown to lead to a downregulation of the corticosterone response, as could be observed in European starlings (*Sturnus vulgaris*) [59]. However, species differences in GC measurements could not be neglected [26], and the results are therefore not necessarily transferable to the white stork.

While comparisons with wild counterparts are often used, their validity is debated, as many behaviours are driven by environmental stimuli rather than internal needs, meaning that the absence of certain wild-type behaviours (like flight) does not necessarily signal poor welfare [21,22,23,60]. Elevated CORTf in wild storks could alternatively reflect exposure to natural stressors such as disease, weather, predation, or food scarcity, all largely absent under human care [44].

In line with Reese et al. (2020) and Haase et al. (2021), we found no significant differences in feather corticosterone levels attributable to the method of flight restraint [2,3]. However, numerically lower logCORTf values were detected in reversibly flight-restrained birds (feather clipping) and in birds that were flight-incapable due to injury. This mirrors findings in great white pelicans (*Pelecanus onocrotalus*), where reversibly deflighted individuals showed a trend towards higher CORTf levels [2]. Methods such as pinioning and feather follicle extirpation are irreversible and surgically invasive; both are ethically controversial and currently prohibited under §6 of the German Animal Welfare Act. These findings underscore the species-specific nature of welfare outcomes in response to flight restraint and indicate that no single method is optimal across taxa [1].

Notably, one of the strongest predictors of CORTf levels was the institutional location (Figure 2). Significant inter-location differences in corticosterone values (ranging from 13.2 ± 0.9 pg/mm to 117.4 ± 26.2 pg/mm) point to the influence of local environmental or management factors, potentially outweighing individual-level effects. Post hoc analysis revealed significant differences in corticosterone levels between several locations. However, no consistent patterns regarding geography, behaviour, or management emerged to explain the variation in corticosterone levels across sites. We attempted to account for as many potential stressors as possible by comparing husbandry practices across institutions and extending the observation period. However, it is likely that not all stressors were captured, as no single specific factor could be identified as being solely responsible. Additionally, as Reese (2020) suggests, ‘even internal factors could play a role. For example, it has been shown that CORT baseline concentrations in in Barn Owls (*Tyto alba*) are genetically correlated.’ [3]. These results parallel differences seen across rehabilitation categories: storks in categories 2 and 3, as well as zoo-housed storks and hand-raised nestlings, showed significantly lower CORTf values compared to wild-living birds (category 1). This suggests that environmental and care conditions have a stronger influence on baseline corticosterone than individual flight capacity. Such results emphasise that welfare cannot be evaluated without considering the wider institutional context.

Behavioural analyses focused on fluttering and locomotion as possible proxies for stress or compensatory movement in flight-restricted birds. Fluttering might indicate an intent to fly or a reaction to social stress, while increased locomotion could serve compensatory functions [61]. To quantify bird movements across different zoological institutions, we calculated a locomotion score: higher scores corresponded to more frequent and faster movements during observation intervals. In univariable models, the locomotion score yielded highly significant *p*-values (*p* < 0.001). However, this variable needed to be removed subsequently from the multivariable model as it correlates strongly with the institution itself. Feeding frequency did not correlate with locomotion. Unlike findings by Haase et al. [2], fluttering was not associated with elevated CORTf. These results suggest that flight restriction does not induce stress-related movement patterns in zoo-housed storks. Though flight in birds fulfils various biological functions, from lung ventilation to thermoregulation [14], many behaviours are regulated externally rather than internally [20], meaning that non-performance of flight does not automatically imply poor welfare [19].

Further behavioural context was offered by the analysis of mixed-species exhibits. Storks housed with species such as the grey crowned crane (*Balearica regulorum*), the western cattle egret (*Ardea ibis*), the helmeted guineafowl (*Numida meleagris*), the great white pelican (*Pelecanus onocrotalus*), or wild duck (*Anas platyrhynchos*) tended to show lower CORTf values. While mixed-species exhibits can increase interspecific aggression, our data show that aggressive behaviours were rare (1.52%), with higher occurrences in single-species settings. Affiliative interactions were roughly four times more frequent (6.56%). These results indicate that social environments in mixed-species settings are not inherently stressful and may even promote positive social interactions. Research supports that mixed-species enclosures, when well designed and species-compatible, offer enrichment and welfare benefits [62,63,64,65]. These dynamics deserve further study in white storks to fully understand potential fitness benefits and social implications. Detailed behavioural distributions across zoos are provided in the Appendix A (Appendix A).

When comparing our analogue and digital behavioural observations, it became apparent that the results substantially overlapped. However, the video recordings were restricted in delivering continuous tracking of the birds. Future studies should ensure this continuity to improve data quality. The posture management survey and several hours of video recordings may not have comprehensively captured all potential stressors affecting the storks, nor all behaviours. For instance, hypothalamic–pituitary–adrenal axis activity could be triggered by uncontrolled external factors such as visitor presence or environmental noise, which were not systematically recorded in this study. Interpretation of the results must consider that some wild storks retained flight ability at sampling and were classified based on their ringing or rehab location, which may not reflect their actual habitat due to potential movement or migration. This limitation does not apply to zoo-housed birds. Moreover, sex, an influencing factor for corticosterone, was not included, as the sampling method (cut feathers) did not allow genetic sexing; required cells at the calamus are only present in plucked feathers [66]. Another potentially relevant factor is age, which was excluded from this analysis. This decision stemmed from the difficulty of obtaining a statistically sufficient sample of nestlings, as young birds have significant blood circulation in the shafts of their growing feathers, complicating corticosterone extraction and analysis. We hypothesised that the season could influence corticosterone levels due to factors such as weather conditions and the breeding or migration period of the birds [67,68]. While season was excluded from statistical analysis due to confounding with sampling time, a visual trend suggested increasing CORTf levels from March to August. In summary, it can be stated that welfare assessments in zoos are applicable but inherently more challenging compared to those in farms or laboratories. This complexity arises from the extensive diversity of species kept, gaps in fundamental biological knowledge, smaller sample sizes, and reduced experimental control [18]. Approaches to assessing animal welfare are multifaceted and remain in need of further refinement [20].

The present study has some limitations that should be acknowledged. First, heterogeneity across study groups in terms of group size, environmental conditions, and observational protocols introduced potential confounding factors despite our efforts to standardise methodologies. Second, behavioural observations, while conducted following rigorous protocols with acceptable inter-observer reliability, inherently contain elements of subjectivity and measurement imprecision that cannot be fully eliminated.

This study employed a non-invasive method of feather corticosterone analysis, allowing for long-term stress assessment without subjecting the birds to acute handling-related stress. Additionally, behavioural observations were conducted in a minimally intrusive manner, preserving naturalistic behaviour. These methodological choices contribute to the refinement of stress monitoring approaches in zoological settings. Moreover, by identifying environmental and husbandry-related factors as significant predictors of corticosterone variation, the findings inform future efforts to design species-appropriate enclosures and practices that may mitigate chronic stress.

## 5. Conclusions

This study emphasised the importance of combining behavioural observations with feather corticosterone feather measurements to comprehensively assess the welfare of white storks. Results indicated that mixed-species exhibits, when carefully designed and tailored to the specific birds’ specific needs, could significantly enhance welfare. Data suggested that zoo and husbandry management might influence CORTf levels more than flight capability, with wild storks exhibiting higher CORTf levels compared to zoo birds, likely due to differing environmental challenges. These findings call for cautious interpretation and encourage future studies to confirm and expand upon these results.

## Figures and Tables

**Figure 1 animals-15-01878-f001:**
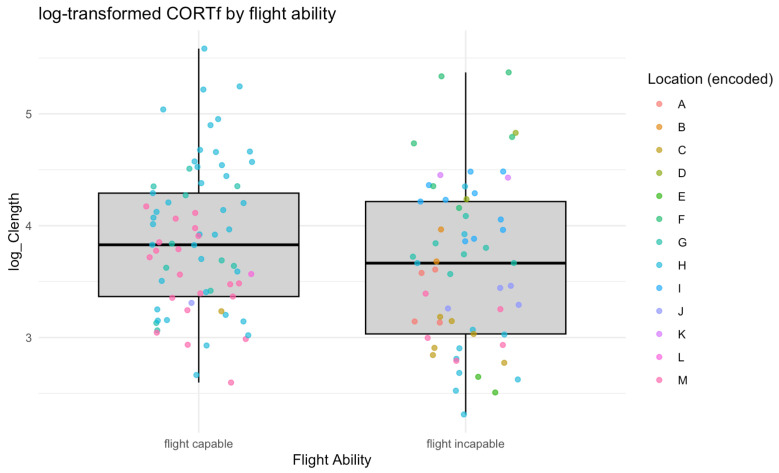
Distribution of log-transformed corticosterone levels of storks with different flight status (flight-capable: zoo nestlings and rehabilitation storks) from various locations (zoos and rehabilitation centres); *n* = 134.

**Figure 2 animals-15-01878-f002:**
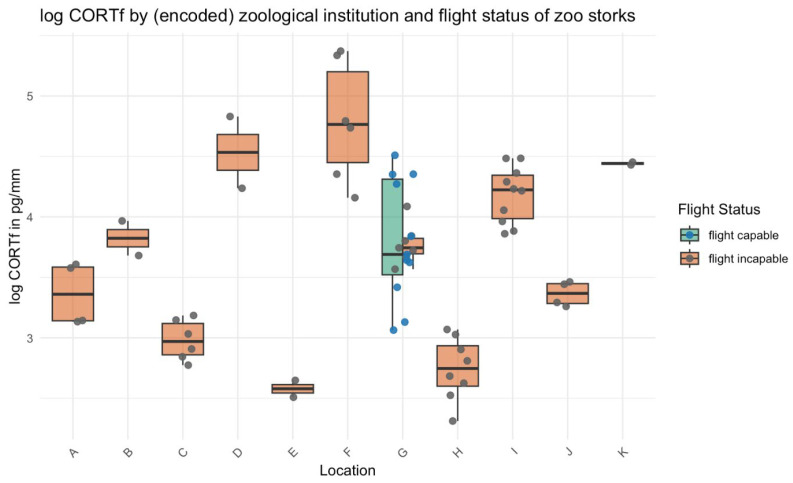
Distribution of log-transformed corticosterone levels in different locations depending on their flight status.

**Figure 3 animals-15-01878-f003:**
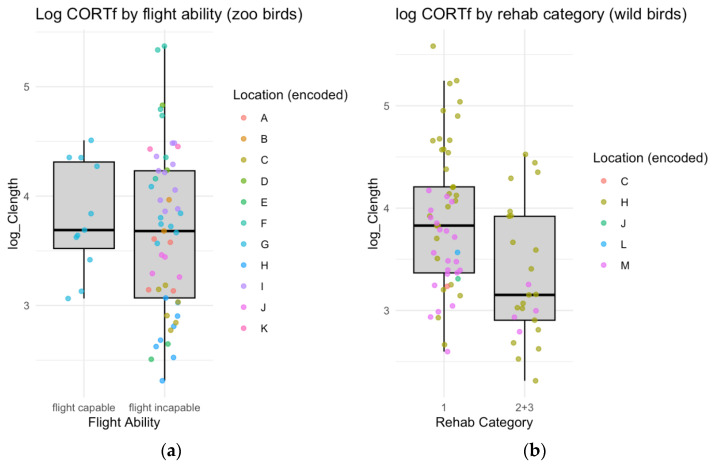
Distribution of log-transformed corticosterone levels in white storks: Comparison between zoological institutions and rehabilitation cases. (**a**) The left panel illustrates corticosterone levels in zoo-housed storks from different zoological institutions, categorized by flight status (flight-capable zoo nestlings and flight incapable adults). (**b**) The right panel shows corticosterone levels in rehabilitated storks, where rehab category 1 includes individuals sampled immediately after their accident, while categories 2 and 3 represent individuals that have been under human care for at least six months.

**Figure 4 animals-15-01878-f004:**
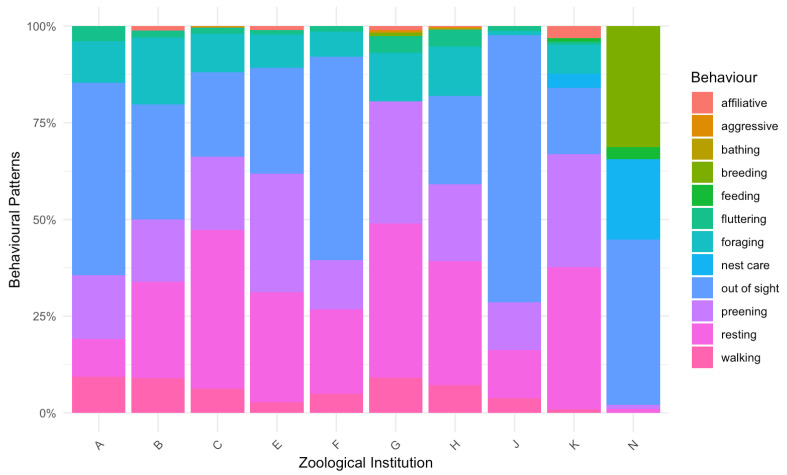
Distribution of different activities of the flocks of white storks within the different zoological institutions evaluated by scan sampling. Zoo N hosted a wild population, a breeding pair was recorded, and G and C had mingled populations.

**Table 1 animals-15-01878-t001:** Sampled individuals per zoological institution.

Zoological Institution	All Individuals	Sampled Individuals	Flight Capable Individuals/Nestlings	Flight Incapable Individuals
A	4	4	0	4
B	2	2	0	2
C ^1^	6	6	x	6
D	2	2	0	2
E	2	2	0	2
F	6	6	0	6
G ^2^	18	7	11	7
H	8	8	0	8
I ^3^	±125	10	100	25
J	4	4	0	4
K	3	2	1	2
Total Number	180	53	112+ x	68

^1,2,3^: Zoological institutions with mixed white stork populations: interaction between resident flight-restricted and wild airworthy individuals.

**Table 2 animals-15-01878-t002:** Ethogram of the white stork.

Behaviour	Definition	Example
Affiliative Behaviour	Interactions between two or more storks promoting bonding.	Up-down display: bill clattering combined with neck movements touching the back. Allopreening: mutual preening in nests, often initiated by females.
Defensive/Aggressive Interactions	Behaviours showing defence or aggression towards others, typically arising from territorial disputes, competition for resources, mate protection, nest defence.	Threat up-down display: neck extension, bill clattering, tail cocking, wing pumping. Hissing: audible hiss combined with defensive displays. Forward stretch display: horizontal posture, retracted neck, pointed bill, aggressive stepping.
Foraging Behaviour	Methods of hunting and prey collection.	Lurking: waiting by animal burrows for prey to appear. Collecting prey: slow walking while picking prey from ground/plants.
Resting and Roosting	Stationary posture characterized by minimal movement, indicating periods of low activity and energy conservation.	Standing: still on one leg and/or laying beak onto the chest.
Locomotion Activity Score	Scale: 0 = resting/roosting, 1 = slow walking, 2 = walking, 3 = fast walking, 4 = accelerating, 5 = running/fluttering.	
Fluttering	Wing flutters or attempts to fly, characterized by short, rapid wing movements.	Lift-off Attempt: abrupt, short wing flaps signalling an initial attempt to fly but not resulting in complete airborne movement.

**Table 3 animals-15-01878-t003:** Univariable linear regression models of 134 white storks; influencing factors (see below) as fixed effects on logCORT; * = significant.

Influencing Factor	Estimate logCORTf in pg/mm	*p*-Value	95%—Confidence Intervall
location L (intercept)	3.568	<0.001 *	2.404; 4.731
location A	−0.202	0.759	−1.503; 1.098
location B	0.256	0.723	−1.169; 1.680
location C	−0.550	0.383	−1.793; 0.694
location D	0.966	0.182	−0.459; 2.391
location E	−0.989	0.172	−2.414; 0.435
location F	1.224	0.056	−0.033; 2.480
location G	0.228	0.706	−0.967; 1.423
location H	0.295	0.621	−0.880; 1.469
location I	0.615	0.320	−0.605; 1.835
location J	−0.214	0.740	−1.488; 1.060
location K	0.874	0.227	−0.551; 2.299
location M	−0.120	0.841	−1.306; 1.066
rehabilitation category 1 (intercept)	3.888	<0.001 *	3.707; 4.069
rehabilitation category 2	−0.675	0.161	−1.622; 0.273
rehabilitation category 3	−0.544	0.001 *	−0.872; −0.215
flight status (capable; intercept)	3.864	<0.001	3.706; 4.022
flight status (incapable)	−0.227	0.058	−0.461; 0.007
study group (reha bird; intercept)	3.821	<0.001	3.657; 3.984
study group (zoo bird)	−0.150	0.234	−0.399; 0.098
study group (zoo nestling)	−0.012	0.956	−0.455; 0.431

**Table 4 animals-15-01878-t004:** Final multivariable linear regression model; fixed effects: location, flight status, rehabilitation category, and study group on logCORTf; * = significant.

Influencing Factor	Estimate logCORTf in pg/mm	*p*-Value	95%—Confidence Intervall
Intercept	3.568	<0.001 *	2.772; 4.363
flight status (incapable)	−0.077	0.743	−0.409; 0.366
rehabilitation category 2	−1.000	<0.001 *	−1.719; −0.555
rehabilitation category 3	−0.328	0.073	−0.758; −0.154
study group (zoo bird)	−1.065	<0.001 *	−1.547; −0.710
study group (zoo nestling)	−1.109	<0.01 *	−1.664; −0.572
location A	−0.097	0.883	−1.180; 0.986
location B	0.360	0.607	−0.793; 1.514
location C	−0.460	0.450	−1.462; 0.542
location D	1.071	0.129	−0.083; 2.224
location E	−0.884	0.209	−2.038; 0.269
location F	1.329	<0.05 *	0.271; 23.87
location G	0.313	0.916	−0.738; 1.364
location H	0.647	0.624	−0.027; 1.592
location I	0.720	0.190	−0.318; 1.758
location J	−0.130	0.255	−1.132; 0.871
location K	0.979	0.831	−0.175; 2.132
location M	−0.052	0.915	−1.029; 0.925

## Data Availability

The data presented in this study was collected and analysed by the authors. The data that support the findings of this study are available on request from the corresponding author.

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
