# Peer review of "Effects of Flight Restraint and Housing Conditions on Feather Corticosterone in White Storks Under Human Care"

_animals, 2025, doi:10.3390/ani15131878_

Round 1
Reviewer 1 Report
Comments and Suggestions for Authors
I would like to thank the Editor for the opportunity to review this detailed manuscript addressing a highly relevant topic. Birds kept in captivity often suffer compromised welfare due to their inability to fly, which can significantly affect both their quality and length of life.
However, I believe several aspects of the manuscript require revision:
-
Title: I recommend shortening the title for clarity and conciseness.
-
Introduction: The introduction should be more concise and focused, with additional references that more robustly support the research topic. It is essential to clearly state the objective of the study or the hypothesis, which is currently only presented in the Materials and Methods section.
-
Materials and Methods: In my opinion, the methodology represents the most critical issue in the manuscript. It remains unclear how the authors accounted for the potential influence of environmental factors on corticosteroid levels. The birds were kept under significantly different conditions and were exposed to stressors for varying durations, which could greatly affect the outcomes.
-
Results: The results are presented appropriately. If feasible, I suggest condensing this section.
-
Discussion: The discussion should be reorganised to rely more on evidence-based conclusions rather than assumptions and speculations. It should also more clearly demonstrate how this study contributes to refining or reducing stress in the sampling process.
Overall, the study addresses an important issue, but improvements in structure and methodology are needed to enhance its scientific value.
Reviewer 2 Report
Comments and Suggestions for Authors
This is an interesting study showing that captive storks do not show higher corticosterone levels than wild individuals. These results are consistent with studies in other bird species. The authors report their findings clearly and discuss them with all required details. I recommend the publication of this work after considering the minor comments and suggestions below.
Lines 100-101: To support your claim, you might be interested in mentioning a recent paper on this topic: Rault et al (2025, Biology Letters).
Lines 126-127: It is worth noting that corticosterone is above all an adaptive response that may be correlated with positive welfare in animals living in enriched environments (e.g., Koolhaas et al 2011, Neurosci Biobehav Rev). It may also be correlated with many other factors, such as the time of the day, the season or the reproductive period.
In Table 1, it would be helpful to have the sum of individuals for each column—would make comparisons easier.
In Table 2, I was a bit surprised to see “out of sight” as part of the ethogram of the storks, as this is not a behavior of the animals per se.
Lines 395-398: Could you show the statistical values for this?
Lines 407-408: Some authors would say that flight is a "behavioral need" for birds, in which case the inability to flight should compromise welfare anyway. I think your data (along with the ones you mention in other bird species) are good empirical evidence against the idea that flight is a behavioral need in storks. Would you say a few words on this question?
Line 526: This acronym should be written in full as it is the only mention in the paper.
Round 2
Reviewer 1 Report
Comments and Suggestions for Authors
I appreciate the authors’ efforts in revising the manuscript in accordance with the reviewer's comments. After reviewing the revised version, I am pleased with the improvements and have no additional suggestions at this time.